# Enhanced Risk Stratification in Early-Stage Endometrial Cancer: Integrating *POLE* through Droplet Digital PCR and L1CAM

**DOI:** 10.3390/cancers15194899

**Published:** 2023-10-09

**Authors:** Seungyeon Joe, Miseon Lee, Jun Kang, Joori Kim, Sook-Hee Hong, Sung Jong Lee, Keun Ho Lee, Ahwon Lee

**Affiliations:** 1Department of Hospital Pathology, College of Medicine, The Catholic University of Korea, Seoul 06591, Republic of Korea; whtmddus0317@naver.com (S.J.); gaon125@naver.com (M.L.);; 2Division of Medical Oncology, Department of Internal Medicine, Seoul St. Mary’s Hospital, College of Medicine, The Catholic University of Korea, Seoul 06591, Republic of Korea; 3Department of Obstetrics and Gynecology, Seoul St. Mary’s Hospital, College of Medicine, The Catholic University of Korea, Seoul 06591, Republic of Korea; 4Cancer Research Institute, College of Medicine, The Catholic University of Korea, Seoul 06591, Republic of Korea

**Keywords:** early-stage endometrial cancer, molecular classification, *POLE*, ddPCR, L1CAM, risk stratification, prognosis

## Abstract

**Simple Summary:**

While a significant number of ECs are successfully treated without recurrence, some cases of EC still result in death even in their early stages. Incorporating molecular classification enhances prognostic accuracy, aiding tailored treatments. This approach has been utilized by the 2021 ESGO/ESTRO/ESP guidelines, the 2022 ESMO guidelines, and the updated 2023 FIGO classification. Our study employed *POLE* ddPCR, a cost-effective and easy-to-perform test, as an alternative to *POLE* NGS for molecular classification. This classification was further enriched by the established prognostic marker, L1CAM, resulting in molecular L1CAM classification. The NSMP group was the largest heterogeneous subgroup. Efforts have been made to find additional markers for further subclassification. When we further categorized the NSMP group, which demonstrates an intermediate prognosis between the *POLEmut*/MMR-D group and the p53abn group, we were able to distinguish the NSMP-L1CAM-positive subgroup, which exhibited a prognosis similar to the p53-mutated subgroup in terms of poorer outcomes. In multivariate analysis, the molecular L1CAM classification showed an independent prognostic factor for recurrence-free survival and overall survival.

**Abstract:**

Aim: In order to enhance risk stratification in early-stage endometrial cancer (EC), we conducted molecular classification using surrogate markers, including the *POLE* droplet digital polymerase chain reaction (ddPCR) and L1CAM immunohistochemistry (IHC). Method: We analyzed archival tumor tissue from 183 early-stage EC patients. *POLE* pathogenic mutations of P286R, V411L, S297F, A456P, and S459F within exons 9, 13, and 14 were detected using a ddPCR, while the mismatch repair (MMR) status was determined by MMR protein IHC and MSI tests. Additionally, we conducted IHC for p53 and L1CAM. Results: The 183 ECs were categorized into four subgroups: *POLE*-mutated (15.9%), MMR-deficient (29.0%), p53-abnormal (8.7%), and non-specific molecular profile (NSMP, 46.4%). We further subcategorized the NSMP subgroup into NSMP-L1CAMneg (41.5%) and NSMP-L1CAMpos (4.9%), which we refer to as the molecular L1CAM classification. The molecular L1CAM classification was an independent prognostic factor for recurrence-free survival (RFS) and overall survival (OS) (*p* < 0.001, each). Conclusion: Integrating molecular L1CAM classification can enhance risk stratification in early-stage EC, providing valuable prognostic information to guide treatment decisions and improve patient outcomes. *POLE* ddPCR might be a cost-effective and easy-to-perform test as an alternative to *POLE* NGS.

## 1. Introduction

Endometrial cancer (EC) is the most common gynecological cancer in developed countries, with 417,367 new cases and 97,370 deaths worldwide in 2020 [1,2,3]. In Korea, the rate of EC has seen a rapid increase recently, with 8.74% annual percent changes [4,5]. Patients diagnosed with early-stage EC exhibit a favorable prognosis. However, 15–20% of patients may suffer from disease recurrence with an aggressive clinical course.

The Cancer Genome Atlas project introduced molecular classification of EC, categorizing ECs into four molecular subtypes: *POLE* exonuclease domain mutation (*POLEmut*) with favorable prognosis; the “hypermutated” subtype, defined by mismatch repair deficiency (MMR-D); the “copy number-high” subtype, with p53-abnormal expression (p53abn) with the poorest prognosis; and the “copy number-low subtype,” also known as “no specific molecular profile” (NSMP) [6,7]. The ProMisE and PORTEC groups subsequently proposed a clinically applicable surrogate marker-based molecular classification system for EC using *POLE* gene sequencing, mismatch repair (MMR) protein immunohistochemistry (IHC) or MSI tests, and p53 IHC [8,9,10,11].

To detect pathogenic mutations in the *POLE* gene, it is recommended to employ next-generation sequencing (NGS) or the Sanger sequencing method. Currently, NGS testing, which encompasses the *POLE* exonuclease domain mutation (EDM, exon 9–14), is available for the identification of all reported pathogenic mutations. However, NGS testing has limitations, including high cost, extended turnaround times, and the need for caution due to the potential for false-positive interpretations. Sanger sequencing for *POLE* generally evaluates exons 9, 11, 13, and 14, where the majority of hotspot mutations are situated. Among these, the five most frequently occurring pathogenic variations, P286R, V411L, S297F, A456P, and S459F, within exons 9, 13, and 14 of the *POLE* EDM collectively account for 84% of known *POLE* pathogenic mutations [12,13,14,15]. The revised WHO classification system and ESMO guideline for EC recommends analyzing these five hotspot mutations using NGS and Sanger sequencing [16,17].

Kim, et al. recently reported that hotspot *POLE* mutations can be detected using droplet digital PCR (ddPCR) assays with high concordance rates (96.7%) compared to Sanger sequencing [18]. In this study, all eight cases with discrepant results exhibited ddPCR positivity and Sanger sequencing negativity. To confirm these findings, *POLE* NGS was conducted, resulting in the detection of *POLE* pathogenic mutations in seven out of eight cases.

In early-stage EC, precise risk assessment is crucial for providing more tailored treatment recommendations and ultimately improving patient outcomes. Clinicopathologic factors for determining risk groups include histologic type, histologic grade, International Federation of Gynecology and Obstetrics (FIGO) stage, myometrial invasion, age, and lymphovascular space invasion (LVSI) [19]. Recently, molecular classification has been integrated into the ESGO/ESTRO/ESP guidelines, ESMO guidelines, and NCCN guidelines [17,20,21]. In the ESGO/ESTRO/ESP and ESMO guidelines, molecular classifications integrated into treatment decisions include adjuvant treatment indications, while NCCN only recommends molecular evaluation for endometrial cancer characterization. In 2023, FIGO staging was updated to encompass the latest advancements in pathological and molecular findings, clinical trial results, prognoses, and survival data. The updated 2023 FIGO staging introduced new subclassifications to incorporate molecular and histological staging systems. Specifically, when molecular classification indicates p53abn or *POLEmut* status in stages I and II, this leads to the disease being upstaged or downstaged (IICm_p53abn_ or IAm*_POLEmut_*) [19].

The L1 cell adhesion molecule (L1CAM; CD171) has been reported as a poor prognostic marker in various solid tumors, including EC [22,23,24,25], colorectal cancer [26], gastric cancer [27,28], ovarian cancer [29], and breast cancer [30]. This poor prognosis is attributed to the ability of L1CAM to enhance cell motility, promoting tumor cell invasion and migration.

In our study of 183 early-stage EC cases, we performed molecular classification using clinically applicable surrogate markers including *POLE* ddPCR. Furthermore, L1CAM immunohistochemical staining was performed to explore its clinical significance, particularly in the context of risk stratification. 

## 2. Materials and Methods

### 2.1. Patients

From January 2013 to December 2018, 183 patients with early-stage EC who underwent surgical treatment at Seoul St. Mary’s Hospital were analyzed. All patients received close follow-up, radiotherapy, chemotherapy, or sequential chemotherapy and radiotherapy based on their risk stratification. Pathological slides of these patients were reviewed by two senior pathologists (Ahwon Lee and Misun Lee) to confirm the pathological parameters, including the histologic type and histologic grade based on the 2020 WHO classification system [16], and tumor staging based on the 2009 FIGO staging system [31] and the updated 2023 FIGO staging system [19]. During the pathological review, representative tumor areas for tissue microarray (TMA) and molecular tests were selected, and cases with insufficient remaining tumor tissue or without formalin-fixed, paraffin-embedded (FFPE) blocks were excluded. This study was approved by the Institutional Review Board of Seoul St. Mary’s Hospital (KC20SISI0979). Investigations were carried out following the rules of the Declaration of Helsinki of 1975, revised in 2013.

### 2.2. Immunohistochemistry for p53, MMR Proteins, and L1CAM

Immunohistochemical staining was conducted using TMA. Two cores of 2 mm diameter tissue were punched out from each representative tumor specimen and arrayed into a recipient block using a manual microarrayer (Quick-Ray set, Unitma, Seoul, Republic of Korea). Each TMA block contained 40 cores of EC.

IHC for p53 and MMR proteins (MLH1, MSH2, MSH6, and PMS2) was performed using primary antibodies for p53 (clone DO7, Ventana Medical Systems, Tucson, AZ, USA), MLH1 (clone M1; Ventana), MSH2 (clone G219-1129; Ventana), MSH6 (clone SP93; Ventana), and PMS2 (clone A16-4; Ventana) in a fully automated manner on a Ventana BenchMark ULTRA platform with an iVIEW DAB detection system (Roche, Mannheim, Germany). The results of the p53 IHC were interpreted as abnormal (mutated-type) staining, characterized by either the strong nuclear expression of tumor nuclei (>80%) or the complete absence of the expression of tumor nuclei (0%), or cytoplasmic staining [32]. If there was no nuclear expression observed in the tumor cells of the TMA, they were re-stained with the whole slide to confirm the complete absence of the expression of tumor nuclei with retained internal control. The results of MMR protein staining were interpreted as abnormal (loss) if any of the following criteria were met: the complete loss of the nuclear expression of both MLH1 and PMS, the loss of both MSH2 and MSH6, the loss of MSH6, or the loss of PMS2. Strong nuclear staining of normal endometrial glands, stromal cells, and lymphoid cells adjacent to the tumor served as the internal positive control. In cases where the MMR protein staining results were equivocal or inconsistent with the MSI-PCR result, the whole slide of the corresponding section was re-stained to verify the MMR protein status.

IHC for L1CAM was performed in line with the manufacturer’s protocol. For L1CAM staining, antigen retrieval was carried out in a pressure cooker (Electric Pressure Cooker CPC-600; Cuisinart, East Windsor, NJ, USA) using 1× citrate buffer (pH 6.0). Sections were incubated with the primary antibody L1CAM (clone 14.10; diluted 1:50; Biolegend, San Diego, CA, USA) at room temperature (22–25 °C) in a humidified chamber. Sections were subsequently incubated with secondary antibodies (EnVision+System-HRP labeled polymer anti-mouse, K4001, DAKO, Glostup, Denmark) at room temperature. The signal of immunoreaction was amplified and revealed using a liquid DAB+ Substrate kit (GBI, Bothell, WA, USA). Subsequently, these slides were counterstained with Harris’s hematoxylin (YD Diagnostics, Yongin, Republic of Korea). If L1CAM immunostaining showed positivity in 10% or more of the tumor cells, it was considered L1CAM-positive. The 10% cut-off was determined based on previous studies [33,34].

### 2.3. Droplet Digital PCR Assay to Detect POLE Mutation

The droplet digital PCR (ddPCR) was performed using a ddPCR system (QX200 Droplet Digital PCR System; Bio-Rad, Hercules, CA, USA) and a commercial kit (Droplex *POLE* Mutation Test, Gencurix, Seoul, Republic of Korea), as per the manufacturer’s recommended protocol [18]. In brief, after thermal cycling, the plate was loaded to measure the endpoint fluorescence signal from each droplet using a Qx200 Droplet Reader (Bio-Rad). The droplet reader was connected to a laptop computer running data analysis QuantaSoft software (v1.6.6.0320; Bio-Rad) and run data analysis was carried out as follows: Each individual droplet was defined on the basis of the fluorescent amplitude as being either positive or negative. The threshold was determined manually based on the amplitude of positive control wells containing wild-type genomic DNA and standard positive DNA. The numbers of positive and negative droplets were distinguished by the threshold, and the given numbers were used for calculating the concentration target in terms of copies/µL. The threshold values of P286R, S297F, V411L, A456P, and S459F were (3000), (3500), (2800), (3200), and (3400), respectively, which were adjusted according to each test condition. According to the fluorescent signal of the negative control, the cut-off for *POLE* mutation was ≥6 copies/20 µL or a mutation index (MI, %, mutant copies for each mutation/the mutant copies of the internal control) ≥ 0.3%.

### 2.4. Microsatellite Instability Test Using PNA Probe-Mediated Real-Time PCR

We performed a microsatellite instability (MSI) test with a U-TOP MSI Detection Kit Plus (Seasun Biomaterials, Daejeon, Republic of Korea), a peptide nucleotide acid (PNA) probe-mediated real-time PCR-based MSI test, as previously described [35]. The U-TOP MSI Detection Kit Plus detects the MSI status by using amplicon melting analysis of five quasi-monomorphic mononucleotide repeat markers (NR21, NR24, NR27, BAT25, and BAT26) and an internal control. Samples with alterations in more than one MSI marker were determined as MSI-H, whereas samples with an alteration in a single MSI marker or no alteration were determined as MSI-L or MSS, respectively. MSI-L and MSS were grouped together for statistical analysis based on a previous report of no significant clinicopathological or molecular differences between MSI-L and MSS colorectal cancers [35].

### 2.5. Statistical Analysis

All of the statistical analyses were performed using SPSS software (version 25.0; IBM Corp., Armonk, NY, USA). The relationship between EC molecular classification and clinicopathological features was analyzed using the chi-square test. The Kaplan–Meier method was used to estimate the recurrence-free survival (RFS) or overall survival (OS), and the differences were compared using the log-rank test. The OS was determined from the pathologic diagnostic date to the last follow-up or the date of the patient’s death, while RFS was defined as the period from the date of primary surgery to the date of cancer recurrence. The prognostic factors were analyzed using Cox’s proportional hazard regression model, including those that were statistically significant in the univariate analysis. For statistical significance, a *p*-value < 0.05 was considered significant.

## 3. Results

### 3.1. Patient Characteristics 

Table 1 shows the clinicopathological characteristics of the 183 patients. The mean age at diagnosis was 55.93 years (range: 30–83 years). The histological subtypes consist of 166 endometrioid carcinoma cases (90.7%) and 17 non-endometrioid carcinoma cases (9.3%; eight serous, one clear cell, one mixed, two dedifferentiated, one mucinous, and four carcinosarcoma). The 2009 FIGO stages at diagnosis were stage IA (133/183 patients, 72.7%), stage IB (33/183 patients, 18.0%), stage II (15/183 patients, 8.2%), and stage III (2/183 patients, 1.1%). The updated 2023 FIGO stages at diagnosis were stage I (135/183 patients, 73.8%) and stage II (48/183 patients, 26.2%).

Patients were risk-stratified (Appendix A) according to the ESMO/ESTRO/ESGO consensus guidelines [36]. The mean follow-up period was 73.5 months (range: 0.3–124.3 months). During the follow-up, 10.9% of the patients experienced disease recurrence and 6.6% of patients died because of their disease.

### 3.2. Molecular Classification Using Surrogate Markers and Its Clinical Significance

Among a cohort of 183 patients with early-stage EC, *POLE* EDM hotspot mutations were found in 29 patients (15.9%). They consisted of P286R (c.857 C > G, exon9) (*n* = 14), V411L (c.1231 G > T/C, exon13) (*n* = 12), A456P (c.1356 G > C, exon14) (*n* = 1), and S459F (c.1376 C > T, exon14) (*n* = 2). MMR protein loss, determined through MMR protein IHC, was found in 52 patients (28.4%). The MSI test revealed forty cases of MSI-H (21.9%), nine cases of MSI-L (4.9%), and one hundred and thirty-four cases of MSS (73.2%). Discrepant cases between the MMR protein IHC and MSI test accounted for a total of 22 cases (IHC normal and MSI-H, five cases; IHC abnormal (loss) and MSI-L or MSS, seventeen cases). We classified cases with MMR protein abnormality (loss) detected by MMR protein IHC, as well as those with MSI-H identified by an MSI test, as MMR-D. The p53 IHC revealed abnormal (mutated) in 17 cases (9.2%). Five “multiple-classifier” cases were identified, exhibiting more than one molecular classifying feature. Molecular classification was prioritized based on the presence of *POLE* mutation > MMR protein loss or MSI-H > p53 abnormality (mutated) [18,20,37]. Four cases in which *POLE* mutation and MMR protein loss by IHC or MSI-H were simultaneously observed using an MSI test were classified as the *POLEmut* subtype. One case in which MMR protein loss by IHC and MSI-H and p53 abnormality (mutated) were simultaneously observed using an MSI test was classified as an MMR-D subtype. The 183 ECs were classified into one of the four molecular subgroups: 29 (15.9%) were *POLEmut*, 53 (29.0%) were MMR-D, 16 (8.7%) were p53abn, and 85 (46.4%) were NSMP ECs.

A comprehensive analysis of the clinical characteristics of molecular classification was conducted (Appendix A). The results revealed significant associations with various factors, including histologic type (*p* < 0.001), histologic grade (*p* < 0.001), updated 2023 FIGO stage (*p* < 0.001), and prognostic risk group (*p* < 0.001) (Appendix A).

The *POLEmut* subtype showed a significant association with the endometrioid histologic subtype (*p* = 0.046). The p53abn subtype exhibited significant associations with old age (*p* = 0.050), the non-endometrioid histologic subtype (*p* < 0.001), and histologically high-grade tumors (*p* < 0.001). The NSMP subtype showed a significant association with the endometrioid histologic subtype (*p* < 0.010) and histologically low-grade tumors (*p* < 0.006). Kaplan–Meier analysis for RFS and OS demonstrated reliable differences between molecular subgroups (Figure 1A,B, *p* < 0.001, each). The five-year RFS and OS for patients with *POLE* mutation were 100% each. Notably, none of the patients with *POLEmut* experienced disease recurrence or succumbed to the disease. Mortality was observed in patients with MMR-D (3.8%, 2/53), p53abn (31.3%, 5/16), and NSMP (5.9%, 5/85). Patients with MMR-D, p53abn, and NSMP ECs showed a five-year RFS of 96.1%, 66.7%, and 85.3%, respectively, and a five-year OS of 96.1%, 73.3%, and 97.5%, respectively.

### 3.3. L1CAM Expression and Its Impact on Prognosis and Molecular Classification

L1CAM was positive in 19 patients (10.4%). According to molecular subgroups, L1CAM overexpression was noted in 3.4% (1/29), 1.9% (1/53), 50.0% (8/16), and 10.6% (9/85) in the *POLEmut*, MMR-D, p53abn, and NSMP subgroups, respectively. L1CAM expression was correlated with old age, the non-endometrioid histologic subtype, and the histological high grade (*p* = 0.001, *p* < 0.001 and *p* < 0.001, respectively).

L1CAM overexpression showed a strong association with worse RFS (five-year RFS 36.5% vs. 94.7%, *p* < 0.001) and OS (five-year OS 71.1% vs. 98.1%, *p* < 0.001) (Figure 2A,B). In the molecular subgroup analysis, L1CAM overexpression was statistically significantly associated with worse RFS and OS (five-year RFS 14.3% vs. 92.5%, *p* < 0.001; five-year OS 75.0% vs. 100.0%, *p* < 0.001, respectively) only in the NSMP subgroup. In the p53abn subgroups, there were trends indicating that L1CAM overexpression might be associated with worse RFS and OS, but these trends did not reach statistical significance.

Building upon these findings and drawing from the previous literature references [23,24,25], we proceeded to subcategorize the NSMP subgroup into the NSMP-L1CAMneg and NSMP-L1CAMpos groups (from now on, we will call it molecular L1CAM classification). Kaplan–Meier analysis for RFS and OS demonstrated reliable differences between the molecular L1CAM subgroups (Figure 1C,D, *p* < 0.001, each). In the molecular L1CAM classification, patients with *POLEmut*, MMR-D, and NSMP-L1CAMneg (158/183, 86.3%) demonstrated favorable outcomes, in contrast to patients with p53abn and NSMP-L1CAMpos (25/183, 13.7%), who had less favorable outcomes (ten-year OS 95.8% vs. 50.3%, *p* < 0.001).

### 3.4. Enhanced Risk Stratification in Early-Stage EC by Integrating Molecular L1CAM Classification

A comprehensive analysis of the clinical characteristics of molecular L1CAM classification was conducted (Table 2). The results revealed significant associations with various factors, including age (*p* = 0.038), histologic type (*p* < 0.001), histologic grade (*p* < 0.001), updated 2023 FIGO stage (*p* < 0.001), prognostic risk group (*p* < 0.001), and adjuvant treatment (*p* = 0.019). Using multivariate analysis, the molecular L1CAM classification was found to be an independent predictor of both RFS and OS (*p* < 0.001, each). Additionally, deep myometrial invasion (> 50%) was associated with worse RFS (*p* < 0.001) and OS (*p* = 0.010) compared to myometrial invasion (<50%) (Table 3).

Considering that p53 abnormality and L1CAM positivity are associated with the poorest prognosis, we divided the patients into two groups: L1CAM/p53-negative, comprising those with both L1CAM negativity and normal p53 (155/183, 84.7%), and L1CAM/p53-positive, including those with either L1CAM positivity or p53 abnormality (28/183, 15.3%). The proportion of L1CAM-positive and p53-abnormal cases according to the molecular L1CAM classification, as well as the distribution of molecular L1CAM subgroups based on L1CAM/p53 categorization, is shown in Figure 3. The L1CAM/p53-positive group exhibited a strong association with worse RFS (five-year RFS 55.4% vs. 95.1%, *p* < 0.001) and OS (five-year OS 77.2% vs. 98.7%; 10-year OS 61.1% vs. 97.8%, *p* < 0.001) compared to L1CAM/p53-negative group (Figure 2C,D).

## 4. Discussion

While a significant number of ECs are successfully treated without recurrence, some cases of EC still result in death even in their early stages. In the 2021 ESGO/ESTRO/ESP guidelines and the recently updated 2022 ESMO guidelines, it is recommended to classify EC into four distinct molecular subgroups to tailor adjuvant therapy [17,20]. In the updated 2023 FIGO staging system, which incorporates prognostic factors into the traditional staging system, the use of molecular classification is encouraged, and if the molecular subtype is known, “m” is added for molecular classification. Especially in early-stage EC, major changes were made including new substages that reflect the molecular alteration of *POLE* and *TP53* genes [21].

In clinical practice, the application of molecular classification for all EC patients is challenging due to medical insurance reasons, which vary widely among countries, as well as economic reasons. Specifically, the *POLE* mutation test poses additional challenges, as it requires repetitive Sanger sequencing or targeted NGS due to the dispersed hotspot regions encompassing exons 9–14. Thus, there are several studies aiming to identify cost-effective testing methods, such as exploring alternative approaches to *POLE* NGS testing or sparing *POLE* testing for selecting cases that might benefit from this analysis [38,39]. Droplet digital PCR can amplify multiple DNA samples using simultaneous reactions in microspheres of several thousand nanoliters, thereby increasing the reliability and sensitivity of the data [40,41,42]. This test has been used in the detection of rare mutations and copy number variations in oncology. The *POLE* ddPCR assay is relatively cost-effective, easy to perform, and has a fast turnaround time compared to NGS testing, with higher sensitivity than Sanger sequencing [18]. A potential limitation of performing ddPCR using FFPE tissue is the susceptibility to droplet classification bias caused by degraded DNA from FFPE samples. Research efforts, including the utilization of machine learning, have been reported to address this issue [43]. According to the present study and Betella et al.’s treatment decision-making algorithm [38], to spare *POLE* testing in clinical practice without impacting treatment decisions in early EC, first conducting IHC for MMR protein, p53, and L1CAM may be suggested. Subsequently, a *POLE* test should be performed only when any of the following criteria are positive: p53-abnormal, L1CAM-positive, stage IB-II, high grade, substantial LVSI. 

On the other hand, it has been reported that positive L1CAM expression was associated with poor prognosis in all patients with EC [23,24,25], in stage I EC [34], and with p53 wild-type EC [25]. Additionally, our study demonstrated positive L1CAM expression as a poor prognostic factor in early-stage EC overall (five-year RFS 36.5% vs. 94.7%, *p* < 0.001; five-year OS 71.1% vs. 98.1%, *p* < 0.001) (Figure 2A,B) and specifically within the NSMP molecular subgroup of early-stage EC (5-year RFS 14.3% vs. 92.5%, *p* < 0.001; 5-year OS 75.0% vs. 100.0%, *p* < 0.001). The NSMP group is the largest heterogeneous subgroup and there have been efforts to find additional markers for further subclassification. When we further categorized the NSMP group, which demonstrates an intermediate prognosis between the *POLEmut*/MMR-D group and the p53abn group, we were able to distinguish the NSMP-L1CAM-positive subgroup, which exhibited a prognosis similar to the p53-mutated subgroup in terms of poorer outcomes (Figure 1). In the 2021 ESGO/ESTRO/ESP guidelines, even stage 1A EC without myometrial invasion is classified as an intermediate risk group if it belongs to the p53abn subtype. EC cases with myometrial invasion from stage I-IVA are categorized as high risk, affecting treatment decisions [20]. Furthermore, in the updated 2023 FIGO staging system, cases of p53abn EC confined to the uterine corpus with any myometrial invasion are classified as Stage IICm_p53abn_ [19]. However, in this study, the NSMP-L1CAM-positive subgroup was limited to nine cases, indicating the need for further validation.

In this study, we intended to detect the five most frequently occurring pathogenic variations within exons 9, 13, and 14 of the *POLE* EDM through ddPCR. These mutations collectively account for 84% of the known *POLE* pathogenic mutations [15]. For the MMR-D subtype classification, we conducted both IHC testing for all four MMR proteins (MLH1, MSH2, MSH6, and PMS2) and MSI testing. A case was classified as MMR-D if either the MMR protein IHC or MSI testing came out positive. In the updated 2023 FIGO staging, MSH6 and PMS2 IHC are recommended as simplified surrogate markers for TCGA molecular classification [19]. The primary use of IHC is recommended because it directly identifies the absent MMR protein(s), while PCR-based MSI tests are not thoroughly validated in non-colorectal cancer, including endometrial cancer [44]. In this study, we found five additional cases categorized into the MMR-D subgroup by additionally performing MSI tests. The diagnostic method to detect MMR-D in EC needs more evaluation. p53 IHC cases with altered expression were classified as the p53abn subtype only if they did not belong to the *POLEmut* or MMR-D subtypes. In our study, MMR proteins and p53 IHC were performed using TMA, which may not fully represent the entire tumor lesion. To address this concern, cases with p53 null expression, equivocal MMR protein IHC results, or discrepancies between the MMR protein IHC and MSI test results were re-examined on whole-slide sections.

Upon classifying early-stage EC into molecular subtypes, the frequencies of the *POLEmut*, MMR-D, p53abn, and NSMP subtypes were found to be 15.9%, 29.0%, 8.7%, and 46.4%, respectively. In the PORTEC cohorts for early-stage EC, their frequencies were 6%, 26%, 9%, and 59%, respectively [32]. A PORTEC cohort study only detected the *POLE* EDM mutation in exons 9 and 13. We additionally detected *POLE* A456P and S459F mutations in exon 14 (*n* = 3).

In the univariate analysis, several factors were found to be statistically significant for worse RFS and OS. These factors include old age, non-endometrial histologic type, high histologic grade, deep myometrial invasion, higher prognostic risk group, higher FIGO stage, and the p53abn/NSMP-L1CAMpos subgroup of molecular L1CAM classification (Table 3). In the multivariate analysis, the molecular L1CAM classification and myometrial invasion were identified as independent prognostic factors for both RFS and OS (Table 3).

There are several limitations in this study: 1) it was a retrospective study, 2) it was conducted in a single institution, and 3) it involved a relatively small sample size, particularly for cases of the NSMP-L1CAMpos subtype. Therefore, further validation through a large-scale, prospective multicenter study is necessary to confirm and generalize our findings.

## 5. Conclusions

In conclusion, our study aimed to enhance risk stratification in early-stage EC by integrating molecular L1CAM classification and *POLE* detection through ddPCR. The molecular L1CAM classification proved to be an independent prognostic factor for both RFS and OS. Despite the limitations of being a single-institution retrospective study with a relatively small sample size, our findings indicate the potential benefit of integrating *POLE* through ddPCR and L1CAM IHC into the current risk-stratification approach. We acknowledge the need for further validation through large-scale, prospective multicenter studies to confirm the utility of our proposed classification method.

## Figures and Tables

**Figure 1 cancers-15-04899-f001:**
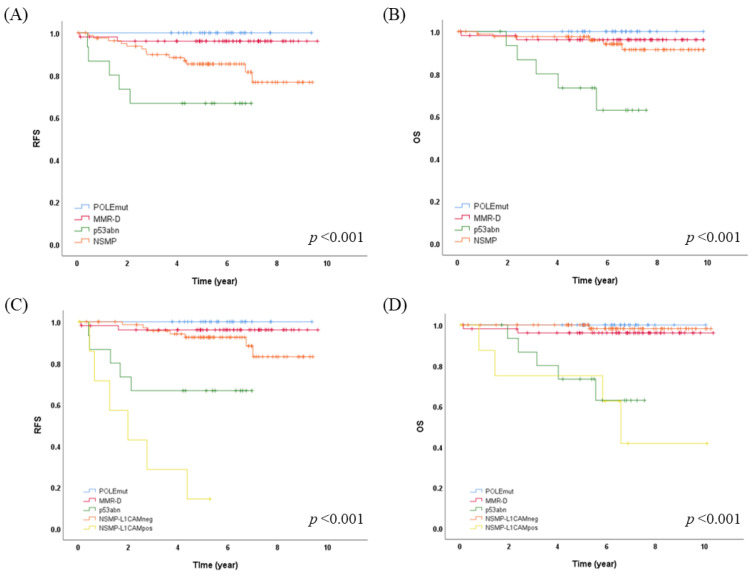
Survival analysis according to the molecular classification and molecular L1CAM classification of early-stage endometrial cancer. Kaplan–Meier curves are shown for the (**A**,**C**) recurrence-free survival and (**B**,**D**) overall survival of 183 patients. *POLEmut*, DNA polymerase epsilon-mutated; MMR-D, mismatch repair-deficient; p53abn, p53-mutated; NSMP, no specific molecular profile.

**Figure 2 cancers-15-04899-f002:**
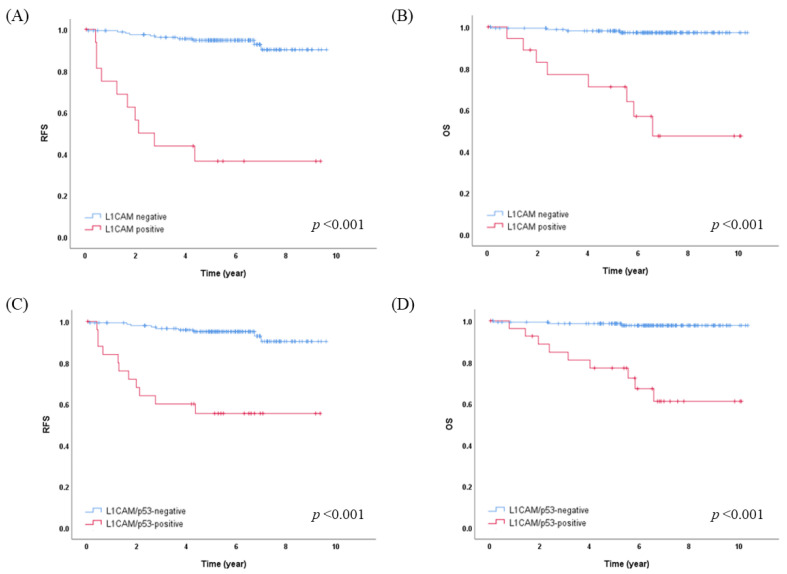
Survival analysis according to L1CAM expression and L1CAM/p53 categorization of early-stage endometrial cancer. Kaplan–Meier curves are shown for the (**A**,**C**) recurrence-free survival and (**B**,**D**) overall survival of 183 patients. L1CAM/p53-negative, L1CAM-negative and p53-normal; L1CAM/p53-positive, either L1CAM-positive or p53-abnormal.

**Figure 3 cancers-15-04899-f003:**
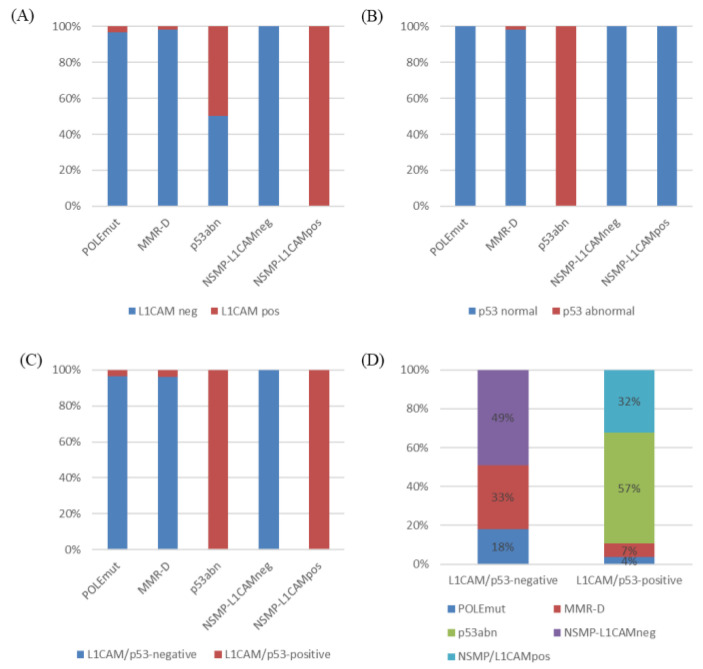
Prevalence of L1CAM-positive (**A**), p53-abnormal (**B**), and L1CAM/p53-positive (**C**) groups according to molecular L1CAM classification. Prevalence of molecular L1CAM classification according to L1CAM/p53 categorization (**D**). L1CAM/p53-negative, L1CAM-negative and p53-normal; L1CAM/p53-positive, either L1CAM-positive or p53-abnormal.

**Table 1 cancers-15-04899-t001:** Patient demographics.

Characteristics	Total
Age	55.93 (30–83)
OP	
Hysterectomy	7 (3.8%)
Hys+BSO	19 (10.4%)
Hys+BSO+LD	157 (85.8%)
Histologic type	
Endometrioid	166 (90.7%)
Non-endometrioid	17 (9.3%)
Histologic grade	
Low	146 (79.8%)
High	37 (20.2%)
LVSI	
Absent	152 (83.1%)
Present	31 (16.9%)
Myometrial invasion	
<50%	145 (79.2%)
>50%	38 (20.8%)
FIGO stage 2009	
IA	133 (72.7%)
IB	33 (18.0%)
II	15 (8.2%)
III	2 (1.1%)
FIGO stage updated 2023	
IA	109 (59.6%)
IB	18 (9.8%)
IC	8 (4.4%)
IIA	9 (4.9%)
IIB	10 (5.5%)
IIC	29 (15.8%)
Prognostic risk group *	
Low	100 (54.6%)
Intermediate	20 (10.9%)
High intermediate	47 (25.7%)
High	16 (8.8%)
Advanced	0 (0.0%)
Adjuvant treatment	
None	147 (80.3%)
Radiotherapy	25 (13.7%)
Chemotherapy	7 (3.8%)
Chemoradiotherapy	4 (2.2%)
Recur/Distant meta	
Absent	160 (87.4%)
Present	20 (10.9%)
NA	3 (1.6%)

BSO, bilateral salpingo-oophorectomy; LD, lymph node dissection; LVSI, lymphovascular space invasion; FIGO, International Federation of Gynecology and Obstetrics; * risk stratified according to the 2016 ESMO/ESTRO/ESGO consensus guidelines.

**Table 2 cancers-15-04899-t002:** Clinicopathological characteristics according to molecular L1CAM classification in early-stage endometrial cancer patients (*n* = 183).

Characteristics	*n* = 183	*POLEmut* *n* = 29	MMR-D *n* = 53	p53abn *n* = 16	NSMP-L1CAM Neg *n* = 76	NSMP-L1CAM Pos *n* = 9	*p*-Value
Age							0.038
<60	134	22 (16.4%)	43 (32.1%)	8 (6.0%)	57 (42.5%)	4 (3.0%)	
≥60	49	7 (14.3%)	10 (20.4%)	8 (16.3%)	19 (38.8%)	5 (10.2%)	
OP							0.521
Hysterectomy	7	0 (0.0%)	2 (28.6%)	1 (14.3%)	3 (42.9%)	1 (14.3%)	
Hys+BSO	19	1 (5.3%)	4 (21.1%)	2 (10.5%)	10 (52.6%)	2 (10.5%)	
Hys+BSO+LD	157	28 (17.8%)	47 (29.9%)	13 (8.3%)	63 (40.1%)	6 (3.8%)	
Histologic type							<0.001
Endometrioid	166	29 (17.5%)	51 (30.7%)	4 (2.4%)	75 (45.2%)	7 (4.2%)	
Non-endometrioid	17	0 (0.0%)	2 (11.8%)	12 (70.6%)	1 (5.9%)	2 (11.8%)	
Histologic grade							<0.001
Low	146	24 (16.4%)	43 (29.5%)	4 (2.7%)	73 (50.0%)	2 (1.4%)	
High	37	5 (13.5%)	10 (27.0%)	12 (32.4%)	3 (8.1%)	7 (18.9%)	
LVSI							0.329
Absent	152	26 (17.1%)	40 (26.3%)	15 (9.9%)	64 (42.1%)	7 (4.6%)	
Present	31	3 (9.7%)	13 (41.9%)	1 (3.2%)	12 (38.7%)	2 (6.5%)	
Myometrial invasion							0.324
<50%	145	24 (16.6%)	40 (27.6%)	14 (9.7%)	62 (42.8%)	5 (3.4%)	
>50%	38	5 (13.2%)	13 (34.2%)	2 (5.3%)	14 (36.8%)	4 (10.5%)	
FIGO stage 2009							0.551
IA	133	23 (17.3%)	36 (27.1%)	14 (10.5%)	56 (42.1%)	4 (3.0%)	
IB	33	5 (15.2%)	11 (33.3%)	0 (0.0%)	13 (39.4%)	4 (12.1%)	
II	15	1 (6.7%)	5 (33.3%)	2 (13.3%)	6 (40.0%)	1 (6.7%)	
III	2	0 (0.0%)	1 (50.0%)	0 (0.0%)	1 (50.0%)	0 (0.0%)	
FIGO stage updated 2023							<0.001
IA	109	20 (18.3%)	29 (26.6%)	4 (3.7%)	55 (50.5%)	1 (0.9%)	
IB	18	2 (11.1%)	6 (33.3%)	0 (0.0%)	10 (55.6%)	0 (0.0%)	
IC	8	0 (0.0%)	2 (25.0%)	2 (25.0%)	2 (25.0%)	2 (25.0%)	
IIA	9	1 (11.1%)	2 (22.2%)	0 (0.0%)	6 (66.7%)	0 (0.0%)	
IIB	10	1 (10.0%)	6 (60.0%)	0 (0.0%)	2 (20.0%)	1 (10.0%)	
IIC	29	5 (17.2%)	8 (27.6%)	10 (34.5%)	1 (3.4%)	5 (17.2%)	
Prognostic risk group *							<0.001
Low	100	20 (20.0%)	28 (28.0%)	3 (3.0%)	48 (48.0%)	1 (1.0%)	
Intermediate	20	3 (15.0%)	6 (30.0%)	2 (10.0%)	7 (35.0%)	2 (10.0%)	
High intermediate	47	6 (12.8%)	17 (36.2%)	1 (2.1%)	20 (42.6%)	3 (6.4%)	
High	16	0 (0.0%)	2 (12.5%)	10 (62.5%)	1 (6.3%)	3 (18.8%)	
Adjuvant treatment							0.019
None	147	25 (17.0%)	41 (27.9%)	12 (8.2%)	64 (43.5%)	5 (3.4%)	
Radiotherapy	25	4 (16.0%)	8 (32.0%)	0 (0.0%)	11 (14.3%)	2 (8.0%)	
Chemotherapy	7	0 (0.0%)	2 (28.6%)	3 (42.9%)	1 (44.0%)	1 (14.3%)	
Chemoradiotherapy	4	0 (0.0%)	2 (50.0%)	1 (25.0%)	0 (0.0%)	1 (25.0%)	

BSO, bilateral salpingo-oophorectomy; LD, lymph node dissection; FIGO, International Federation of Gynecology and Obstetrics; LVSI, lymphovascular space invasion; MMR-D, mismatch repair-deficient; p53abn, p53-mutated; NSMP, no specific molecular profile; *POLEmut*, DNA polymerase epsilon-mutated; * risk stratified according to the 2016 ESMO/ESTRO/ESGO consensus guidelines.

**Table 3 cancers-15-04899-t003:** Univariate and multivariate Cox regression analyses of prognostic variables for recurrence-free survival and overall survival in 183 patients with early-stage endometrial cancer.

	RFS				OS			
	Univariate Analysis		Multivariate Analysis		Univariate Analysis		Multivariate Analysis	
Parameters	Hazard Ratio (95% CI)	*p*-Value	Hazard Ratio (95% CI)	*p*-Value	Hazard Ratio (95% CI)	*p*-Value	Hazard Ratio (95% CI)	*p*-Value
age (<60 vs >60 years)	4.336 (1.772–10.611)	0.001	-		6.353 (1.911–21.122)	0.003	-	
Histologic type (endometrioid vs non-endometrioid)	6.309 (2.410–16.511)	<0.001	-		13.836 (4.424–43.269)	<0.001	-	
Histologic grade (grade 1, 2 vs grade 3, high grade)	4.978 (2.067–11.989)	<0.001	-		14.322 (3.865–53.078)	<0.001	-	
Myometrial invasion (<50% vs >50%)	3.330 (1.375–8.063)	0.008	3.845 (1.568–9.428)	<0.001	0.264 (0.0.85–0.821)	0.021	4.535 (1.443–14.251)	0.010
Prognostic risk group								
intermediate	5.604 (1.448–21.693)	0.013	-		8.853 (0.989–79.213)	0.051	-	
high	10.645 (2.927–38.720)	<0.001	-		21.700 (2.668–176.469)	0.004	-	
Updated 2023 FIGO stage (stage 1 vs stage 2)	5.882 (2.344–14.756)	<0.001	-		4.248 (1.348–13.389)	0.014	-	
Molecular L1CAM classification (*POLEmut*, MMR-D, NSMP-L1CAMneg vs p53abn, NSMP-L1CAMpos)	13.537 (5.401–33.932)	<0.001	15.005 (5.883–38.269)	<0.001	22.585 (6.104–83.560)	<0.001	24.807 (6.669–92.277)	<0.001

RFS, recurrence free survival; OS, overall survival; FIGO, International Federation of Gynecology and Obstetrics; *POLEmut*, DNA polymerase epsilon-mutated; MMR-D, mismatch repair-deficient; p53abn, p53-mutated; NSMP, no specific molecular profile; HR, hazard ratio; Cl, confidence interval.

## Data Availability

The data presented in this study are available on request from the corresponding author. The data are not publicly available due to ethical restrictions.

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
