# Peer review of "Enhanced Risk Stratification in Early-Stage Endometrial Cancer: Integrating POLE through Droplet Digital PCR and L1CAM"

_cancers, 2023, doi:10.3390/cancers15194899_

Round 1
Reviewer 1 Report
Dear Authors,
It was a pleasure to review your manuscript entitled: “Enhanced risk stratification in early-stage endometrial cancer: Integrating Molucular-L1CAM Classification and Simplified L1CAM/p53 Categorization".
Please, find below the comments that I would ask you to address.
General:
· All studies conducted to validate the molecular classification for the prognostic purpose include POLE evaluation, which is required also according to WHO guidelines in order to classify an endometrial cancer. You suggest to simplify the molecular classification by omitting POLE evaluation and adding L1CAM evaluation. This suggestion presents many limitations. In this study, there were not POLE+p53abn multiple classifiers: how would you consider these cases? They would be misclassified according with the simplified evaluation you suggest. Recently, Promise and Transportec groups evaluate how to refine the prognosis in NSMP molecular subgroup and they found that ER expression correlate with prognosis, while L1CAM do not. How can your results be commented in light of theirs?
Introduction:
· Line 56 “only”: Actually european and NCCN guidelines recommend adjuvant treatment also in intermediate risk (i.e. vaginal brachitherapy)
· Line 63 “MSI test”: Generally accepted as an alternative method for MMR evaluation, but not considered one of the three surrogate markers from the two validation studies (Promise and Trasportec)
· Line 76 “therapy avoiding overtreatment”: consider to reword, not clear
· Line 79, ref 26: These guidelines have been reviewed in 2022. Consider to update bibliography
· Line 79-82: Not only ESGO guidelines have been implemented with molecular classes, but ESMO and NCCN guidelines as well. ESMO and ESGO guidelines base adjuvant treatment indications on molecular evaluation, while NCCN just recommends molecular evaluation for endometrial cancer caracterization
Methods:
· Line 103 “represented”: do you mean representative?
· Line 110 “Two cores of 2 mm di-110 ameter tissue were punched out”: might this justify the low frequency of p53 abn cases?
· Line 119-121: What about the rare cytoplasmatic pattern? Was it considered as abnormal?
· Line 143-169: you used two different methods to evaluate POLE mutations: Sanger and NGS. Were all cases analyzed with both methods? What was the concordance rate? Did you have any positive control for the test or are these methods been previously validated?
Results:
· Line 203: I would cite either ESMO 2022 guidelines or ESGO/ESTRO/ESP 2021 guidelines and be consistent with this choice when you attribute the cases to a risk class.
· Line 205-206: This is a comment, not a result.
· Line 220-221: I suggest to rephrase: IHC negative and positive might be substitute with normal/abnormal for more clarity
· Line 221-223: What have you done in equivocal cases?
· Line 223: Consider to substitute abnormal instead of positive for more clarity
· Line 223-226: The percentage of multiple classifiers is quite low compared to other papers, as well as the frequency of p53 abn ECs
· Line 227: verify the number
· Line 298-300: Is it part of the results?
Discussion:
· Line 318: Not only POLE and p53 are responsible for substantial changes in new FIGO staging system... The changes are substantial even without molecular evaluation, based on the introduction of prognostic factors in the traditional staging system based on the spatial distribution of cancer.
· Line 326-29: this concept is not clear, a rewording is required
· Line 331-332: better explain what do you mean for elimination of the intermediate risk group
· Line 338: Expand the concept. Not clear how it is relevant within this study
· Line 347-348: There are many studies about this topic and cost-effectiveness evaluation has already clarified that IHC evaluation is sufficiently reliable in ECs. Actually, it's more frequent having MS stable with deficient MMR.
· Line 351: Not cited in the result: have you found any subclonal expression of p53?
· Line 352: Have you considered the prognostic outcome to decide how to classify EC with positivity of multiple surrogate markers?
· 354-355: Which results did you obtain on the whole slide? What was the concordance rate?
· Line 364: see comment about tables
· Line 365-369: not clear
· Line 370: Have you esteemed the cost-effectiveness of doing the analysis omitting POLE evaluation? POLE can be evaluated using PCR, with lower cost than NGS. Furthermore, many studies evaluate how to spare POLE testing selecting cases which might benefit from this analysis (for example, Betella I, Fumagalli C, Rafaniello Raviele P, et al. Int J Gynecol Cancer. 2022. doi:10.1136/ijgc-2022-003480 and Imboden S, Nastic D, Ghaderi M, et al. Implementation of the 2021 molecular ESGO/ESTRO/ESP risk groups in endometrial cancer. Gynecol Oncol 2021;162:394–400.)
· Line 383-387: What about technical issue related to POLE analysis? Do you think this might be a limitation?
Tables and figures:
· Table 1:
o FIGO stage 2009: What is stage IIA and IIB? Is it not all stage II according with 2009 FIGO stage?
o Prognostic risk group: I would specify if you used risk classification system with or without considering the molecular class
o BD, Bilateral salpingo-oophorectomy: not commonly used abbreviation
· Table 2:
o From the statistical point of view, what is the practical meaning of correlations with other prognostic classes in univariate/multivariate analysis? I would correlate only with histopathological features: the risk stratification group are already done correlating various risk factors
· Figures
o Consider to use the same time period (for example 10 years) in both the RFS and OS curves. It would be more intuitive if the proportion of x and y axis are maintained identical.
See above
Author Response
Reviewer 1.
General:
- All studies conducted to validate the molecular classification for the prognostic purpose include POLE evaluation, which is required also according to WHO guidelines in order to classify an endometrial cancer. You suggest to simplify the molecular classification by omitting POLE evaluation and adding L1CAM evaluation. This suggestion presents many limitations. In this study, there were not POLE+p53abn multiple classifiers: how would you consider these cases? They would be misclassified according with the simplified evaluation you suggest. Recently, Promise and Transportec groups evaluate how to refine the prognosis in NSMP molecular subgroup and they found that ER expression correlate with prognosis, while L1CAM do not. How can your results be commented in light of theirs?
In our study, we overlooked the possibility of overtreating POLE+p53abn EC patients, as we did not have any cases with the POLE+p53abn dual classifier. We have removed all content related to L1CAM/p53 categorization. Instead, we have emphasized in our study the detection of POLE mutations using ddPCR and the enhancement of risk stratification when adding L1CAM to molecular classification. During this process, we excluded 64 cases that had undergone POLE Sanger sequencing, resulting in some changes in the numbers in our results.
Based on the results of the L1CAM/p53 categorization, we continue to consider L1CAM positivity as a poor prognostic factor, on par with p53 abnormalities. We fully acknowledge that omitting POLE evaluation in high-grade, high-stage cases contradicts all current guidelines. Therefore, we have included the following sentence in the discussion section.
In Line 341-346 “According to the present study and Betella et al.'s treatment decision-making algorithm [38], to spare POLE testing in clinical practice without impacting treatment decisions in early EC, it might be suggestive to first conduct IHC for MMR protein, p53, and L1CAM. Subsequently, perform a POLE test only when any one of the following criteria is positive: p53 abnormal, L1CAM positive, stage IB-II, high grade, substantial LVSI.”
However, in our study, we did not conduct immunostaining for ER (Estrogen Receptor), so we were unable to mention the relative importance of ER and L1CAM.
Introduction:
- Line 56 “only”: Actually european and NCCN guidelines recommend adjuvant treatment also in intermediate risk (i.e. vaginal brachitherapy)
Line 50: I agree with the opinion, and the sentence has been deleted.
- Line 63 “MSI test”: Generally accepted as an alternative method for MMR evaluation, but not considered one of the three surrogate markers from the two validation studies (Promise and Trasportec)
Line 60: We conducted both tests and considered the result positive if either of them yielded a positive outcome. While The ProMisE and PORTEC groups have acknowledged MMRd IHC as a surrogate marker, we believe that further evaluation is needed to establish one gold standard test between MSI testing and MMRd IHC. Also in ESMO MMR-IHC and MSI assay both are available to diagnostic test.
- Line 76 “therapy avoiding overtreatment”: consider to reword, not clear
Line 80: I agree with the opinion, and the sentence has been deleted.
- Line 79, ref 26: These guidelines have been reviewed in 2022. Consider to update bibliography
Line 83: We update bibliography
- Line 79-82: Not only ESGO guidelines have been implemented with molecular classes, but ESMO and NCCN guidelines as well. ESMO and ESGO guidelines base adjuvant treatment indications on molecular evaluation, while NCCN just recommends molecular evaluation for endometrial cancer caracterization
Line84-87: We have incorporated the additional information you kindly provided.
Methods:
- Line 103 “represented”: do you mean representative?
Line 112: We corrected type-error.
- Line 110 “Two cores of 2 mm diameter tissue were punched out”: might this justify the low frequency of p53 abn cases?
Line 119: We also recognized the possibility that research using TMA may not fully represent whole slide results. To address this, we conducted a reevaluation of cases demonstrating complete negative p53 in the entire slide.
Line 130-133“If there was no nuclear expression observed in the tumor cells of the TMA, they were re-stained with the whole slide to confirm the complete absence of the expression of tumor nuclei with retained internal control.”
Line 386-387: Furthermore, “In the PORTEC cohorts for early-stage EC, their frequency of p53 mutated type was 9% [38]”, similar to our result (8.7%).
- Line 119-121: What about the rare cytoplasmatic pattern? Was it considered as abnormal?
: We have made the following modifications to the interpretation criteria.
Line 128-130:“The results of p53 IHC were interpreted as abnormal (mutated type) staining, characterized by either the strong nuclear expression of tumor nuclei (>80%) or the complete absence of the expression of tumor nuclei (0%) or cytoplasmic staining [32].”
In cases of cytoplasmic positivity, we interpreted them as p53 abnormal. However, in present study, cytoplasmic positivity was observed concurrently with nuclear positivity.
- Line 143-169: you used two different methods to evaluate POLE mutations: Sanger and NGS. Were all cases analyzed with both methods? What was the concordance rate? Did you have any positive control for the test or are these methods been previously validated?
Line 153: During this revision process, we conducted a re-analysis using 183 cases where POLE ddPCR was performed, excluding cases that underwent Sanger sequencing only. The utility of POLE ddPCR in this study was adequately validated through comparative testing with Sanger sequencing and NGS in a previous publication [Kim, et al., ref 18]. We conducted experiments using the same methodology and equipment as described in Kim's paper.
Results:
- Line 203: I would cite either ESMO 2022 guidelines or ESGO/ESTRO/ESP 2021 guidelines and be consistent with this choice when you attribute the cases to a risk class.
Line 203: "The reason we conducted risk stratification based on the 2016 ESMO/ESTRO/ESGO guidelines was to compare it with clinical risk stratification before the integration of molecular classification and our hospital use 2016 ESMO/ESTRO/ESGO guidline in practice. However, we will make the necessary revisions to ensure consistency with the updated prognostic risk stratification, as it is essential for our study.”
- Line 205-206: This is a comment, not a result.
Line 203-204: We have revised the sentence as follows ;
“Patients were risk stratified (Figure. S1) according to the ESMO/ESTRO/ESGO consensus guidelines [36].”
Line 206: We deleted the following sentence.
“Kaplan–Meier survival analysis demonstrated reliable differences between clinical risk groups”
- Line 220-221: I suggest to rephrase: IHC negative and positive might be substitute with normal/abnormal for more clarity
Line220: We corrected as you suggested.
- Line 221-223: What have you done in equivocal cases?
Line 137-140: “In cases where the MMR protein staining results were equivocal or inconsistent with the MSI-PCR result, the whole slide of the corresponding section was re-stained to verify the MMR protein status.”
Line 221-222: “We classified cases with MMR protein abnormal (loss) detected by MMR protein IHC, as well as those with MSI-H identified by an MSI test, as MMR-D.”
- Line 223: Consider to substitute abnormal instead of positive for more clarity
Line 221: We substituted abnormal (loss) instead of positive for more clarity.
- Line 223-226: The percentage of multiple classifiers is quite low compared to other papers, as well as the frequency of p53 abn ECs
Multiple classifier identified 2.7% (5/183) in our study, which is similar to previous reports (3.0% (107/3518) ref 37; 2.08% (5/240) ref 18).
Line 223-226: “Five ‘multiple- classifier’ cases were identified, exhibiting more than one molecular classifying feature. Molecular classification was prioritized based on the presence of POLE mutation > MMR protein loss or MSI-H > p53 abnormal (mutated) [18, 37].”
Line 386-387: In the PORTEC cohorts for early-stage EC, their frequency of p53 mutated type was 9% [32]”, similar to our result (8.5%).
Line 384-387: “Upon classifying early-stage EC into molecular subtypes, the frequencies of the POLEmut, MMR-D, p53abn, and NSMP subtypes were found to be 15.9%, 29.0%, 8.7%, and 46.4%, respectively. In the PORTEC cohorts for early-stage EC, their frequencies were 6%, 26%, 9%, and 59%, respectively [32].”
- Line 227: verify the number
Line 230: We corrected the numbers in this sentence.
- Line 298-300: Is it part of the results?
Line 301: We deleted the sentence, since it is not part of the result.
Discussion:
- Line 318: Not only POLE and p53 are responsible for substantial changes in new FIGO staging system... The changes are substantial even without molecular evaluation, based on the introduction of prognostic factors in the traditional staging system based on the spatial distribution of cancer.
: We agree with your opinion, we followed your suggestions
Line 320-322: “In the updated 2023 FIGO staging system, which incorportates prognostic factors into the traditional staging system, the use of molecular classification is encouraged,”
- Line 326-29: this concept is not clear, a rewording is required and Line 331-332: better explain what do you mean for elimination of the intermediate risk group
:We agree with your opinion. The entire sentence has been deleted, and we have reinterpreted the results accordingly.
Line 353-365“The NSMP is the largest heterogeneous subgroup and there have been efforts to find addi-tional markers for further subclassification. When we further categorized the NSMP group, which demonstrates an intermediate prognosis between the POLEmut/MMR-D group and the p53abn group, we were able to distinguish the NSMP-L1CAM positive subgroup, which exhibited a prognosis similar to the p53 mutated subgroup in terms of poorer out-comes (Figure 1). In the 2021 ESGO/ESTRO/ESP guideline, even stage 1A EC without myometrial invasion is classified as an intermediate risk group if it belongs to the p53abn subtype. For EC cases with myometrial invasion from stage I-IVA, they are categorized as high risk, affecting treatment decisions [20]. Furthermore, in the updated 2023 FIGO stag-ing system, cases of p53abn EC confined to the uterine corpus with any myometrial inva-sion are classified as Stage IICmp53abn [19]. However, in this study, the NSMP-L1CAM posi-tive subgroup was limited to 9 cases, indicating the need for further validation.”
- Line 338: Expand the concept. Not clear how it is relevant within this study
Line 61-64: We have relocated the following sentence to the introduction section with some modifications:
Line 64-72: “However, NGS testing has limitations, including high cost, extended turnaround times, and the need for caution due to the potential for false-positive interpretations. Sanger se-quencing for POLE generally evaluates exons 9, 11, 13, and 14, where the majority of hotspot mutations are situated. Among these, the five most frequently occurring patho-genic variations, P286R, V411L, S297F, A456P, and S459F, within exons 9, 13, and 14 of the POLE EDM collectively account for 84% of known POLE pathogenic mutations [12-15]. The revised WHO classification system and ESMO guideline for EC recommends analyz-ing these 5 hotspot mutations using NGS and Sanger sequencing [16,17].”
- Line 347-348: There are many studies about this topic and cost-effectiveness evaluation has already clarified that IHC evaluation is sufficiently reliable in ECs. Actually, it's more frequent having MS stable with deficient MMR.
: We have made the following modifications.
Line 373-377: The primary use of IHC is recommended because it directly identifies the absent MMR protein(s), while PCR-based MSI tests are not thoroughly validated in non-colorectal cancer including endometrial cancer [44]. In this study, we found five additional cases categorized into the MMR-D subgroup by additionally performing MSI tests. The diagnostic method to detect MMR-D in EC need more evaluation.
- Line 351: Not cited in the result: have you found any subclonal expression of p53?
: “This classification strategy for multiple classifiers was based on various factors, including morphologic features, the frequent subclonal overexpression of p53, hierarchical cluster-ing study with TCGA data, and clinical outcomes [35].” The intention of this sentence was to interpret the multiple classifiers based on reference [35], not our experimental results. Ultimately, during the process of result reinterpretation, this entire sentence has been deleted.
- Line 352: Have you considered the prognostic outcome to decide how to classify EC with positivity of multiple surrogate markers?
We performed molecular classification for ‘multiple classifier’, applying the method described in most comprehensive paper on multiple classifier research (also this same method is recommended by the 2021 ESGO/ESTRO/ESP guidelines).
Line 223-226: Five ‘multiple- classifier’ cases were identified, exhibiting more than one molecular classifying feature. Molecular classification was prioritized based on the presence of POLE mutation > MMR protein loss or MSI-H > p53 abnormal (mutated) [18,20,37].
- Concin, N.; Matias-Guiu, X.; Vergote, I.; Cibula, D.; Mirza, M.R.; Marnitz, S.; Ledermann, J.; Bosse, T.; Chargari, C.; Fagotti, A.; et al. ESGO/ESTRO/ESP guidelines for the management of patients with endometrial carcinoma. Int J Gynecol Cancer 2021, 31, 12-39.
- Leon-Castillo, A.; Gilvazquez, E.; Nout, R.; Smit, V.T.; McAlpine, J.N.; McConechy, M.; Kommoss, S.; Brucker, S.Y.; Carlson, J.W.; Epstein, E.; et al. Clinicopathological and molecular characterisation of 'multiple-classifier' endometrial carcinomas. J Pathol 2020, 250, 312-322.
- Line 354-355: Which results did you obtain on the whole slide? What was the concordance rate?
Line 379-383: “In our study, MMR proteins and p53 IHC were performed using TMA, which may not fully represent the entire tumor lesion. To address this concern, cases with p53 null expression, equivocal MMR protein IHC results, or discrepancies between the MMR protein IHC and MSI test results were re-examined on whole-slide sections.”
We did not perform the concordance evaluation between TMA and whole slide results.
- Line 364: see comment about tables
- Line 365-369: not clear
:I have revised the sentence for clarity.
Line 397-399: “There are several limitations in this study: 1) it was a retrospective study, 2) con-ducted in a single institution, and 3) involved a relatively small sample size, particularly for cases of the NSMP-L1CAMpos subtype. Therefore, further validation through a large-scale, prospective multicenter study is necessary to confirm and generalize our findings.”
- Line 370: Have you esteemed the cost-effectiveness of doing the analysis omitting POLE evaluation? POLE can be evaluated using PCR, with lower cost than NGS. Furthermore, many studies evaluate how to spare POLE testing selecting cases which might benefit from this analysis (for example, Betella I, Fumagalli C, Rafaniello Raviele P, et al. Int J Gynecol Cancer. 2022. doi:10.1136/ijgc-2022-003480 and Imboden S, Nastic D, Ghaderi M, et al. Implementation of the 2021 molecular ESGO/ESTRO/ESP risk groups in endometrial cancer. Gynecol Oncol 2021;162:394–400.)
Line 330 : We intended to assess the cost-effectiveness of POLE ddPCR. However, since it has not yet been officially released by the company as a diagnostic product, we encountered difficulties in comparing prices. Nevertheless, it is anticipated to be more affordable than NGS testing and more expensive than Sanger sequencing.
- Line 383-387: What about technical issue related to POLE analysis? Do you think this might be a limitation?
We added following sentence.
Line 333-341: “Droplet digital PCR can amplify multiple DNA samples using simultaneous reactions in microspheres of several thousand nanoliters, thereby increasing reliability and sensitivity of the data [40-42]. This test has been used in detection of rare mutations and copy number variations in oncology field. The POLE ddPCR assay is relatively cost-effective, easy to perform, and a fast turnaround time compared to NGS testing with higher sensitivity than sanger sequencing [18]. A potential limitation of performing ddPCR using FFPE tissue is the susceptibility to droplet classification bias caused by degraded DNA from FFPE samples. Research efforts, including the utilization of machine learning, have been reported to address this issue [43].”
Tables and figures:
- Table 1:
o FIGO stage 2009: What is stage IIA and IIB? Is it not all stage II according with 2009 FIGO stage?
: Since our study cases collected according to 2023 FIGO, two cases with low grade tumor involved only the uterus and ovary (1A by 2023 FIGO) were classified as FIGO stage 3 according to the 2009 FIGO classification. We have adjusted our classification accordingly.
o Prognostic risk group: I would specify if you used risk classification system with or without considering the molecular class
: We have included the year in the guidelines we used to emphasize that it was a concept that did not include molecular classification.
Table 1 : “*Risk stratified according to the 2016 ESMO/ESTRO/ESGO consensus guideline."
o BD, Bilateral salpingo-oophorectomy: not commonly used abbreviation
: We corrected as BSO.
- Table 2:
o From the statistical point of view, what is the practical meaning of correlations with other prognostic classes in univariate/multivariate analysis? I would correlate only with histopathological features: the risk stratification group are already done correlating various risk factors
: We conducted univariate analysis on histopathologic and prognostic factors, selecting statistically significant variables, and subsequently proceeded with multivariate analysis. For instance, LVI was excluded from the analysis as it did not demonstrate significance in the univariate analysis.
- Figures
o Consider to use the same time period (for example 10 years) in both the RFS and OS curves. It would be more intuitive if the proportion of x and y axis are maintained identical.
: We corrected as you reccommended. We used the same time period (11 years) in both the RFS and OS curves
Reviewer 2 Report
This study evaluated the role of L1CAM marker in EC prognosis. The authors report that L1CAM could be used to further subcategorise NSMP subgroup of EC patients. I have the following comments:
1. Is this a retrospective analysis of prospectively collected data or a retrospective study?
2. How many patients with EC are treated in the authors’ institution each year?
3. Were all consecutive patients included?
4. What were the exclusion criteria?
5. In the discussion section the authors should further discuss a possible clinical implications of this study. Is this sample big enough to draw any definite conclusions?
Author Response
Reviewer 2.
This study evaluated the role of L1CAM marker in EC prognosis. The authors report that L1CAM could be used to further subcategorise NSMP subgroup of EC patients. I have the following comments:
- Is this a retrospective analysis of prospectively collected data or a retrospective study?
: Our study is a retrospective study
Line 397-399: “There are several limitations in this study: 1) it was a retrospective study, 2) conducted in a single institution, and 3) involved a relatively small sample size, particularly for cases of the NSMP-L1CAMpos subtype.”
- How many patients with EC are treated in the authors’ institution each year?
: Currently, the incidence of endometrial cancer has significantly increased. However, during the study period (2013 to 2018), an average of approximately 60 to 70 patients with endometrial cancer underwent surgery per year.
- & 4. Were all consecutive patients included? What were the exclusion criteria?
: We initiated the study with a cohort comprising all consecutive patients who underwent hysterectomy for endometrial cancer during the period from 2013 to 2018. To narrow down the focus to early-stage cancer, we excluded 115 cases with advanced-stage cancer. Furthermore, 73 cases were excluded as they had only small residual lesions in the uterine specimens after curettage. Consequently, initially, we included a total of 247 patients in the study. However, during the revision process, we restricted the study population to cases where POLE ddPCR was conducted, excluding 64 cases subjected to Sanger sequencing.
- In the discussion section the authors should further discuss a possible clinical implications of this study. Is this sample big enough to draw any definite conclusions?
: During the revision process, we re-evaluated our results in response to the valuable comments from the reviewers. We also conducted additional discussions regarding the potential clinical implications of our findings. We acknowledge that the sample size of our study is not enough, and we have highlighted this as one of the limitations of our study.
: Line 326-346: “In clinical practice, the application of molecular classification for all EC patients is challenging due to medical insurance reasons, which vary widely among countries, and also economic reasons. Specifically, the POLE mutation test poses additional challenges, as it requires repetitive Sanger sequencing or targeted NGS due to the dispersed hotspot regions encompassing exons 9-14. Thus, there are several studies aiming to identify cost-effective testing methods, such as exploring alternative approaches to POLE NGS testing or sparing POLE testing for selecting cases that might benefit from this analysis [38,39]. Droplet digital PCR can amplify multiple DNA samples using simultaneous reac-tions in microspheres of several thousand nanoliters, thereby increasing reliability and sensitivity of the data [40-42]. This test has been used in detection of rare mutations and copy number variations in oncology field. The POLE ddPCR assay is relatively cost-effective, easy to perform, and a fast turnaround time compared to NGS testing with higher sensitivity than sanger sequencing [18]. A potential limitation of performing ddPCR using FFPE tissue is the susceptibility to droplet classification bias caused by de-graded DNA from FFPE samples. Research efforts, including the utilization of machine learning, have been reported to address this issue [43]. According to the present study and Betella et al.'s treatment decision-making algorithm [38], to spare POLE testing in clinical practice without impacting treatment decisions in early EC, it might be suggestive to first conduct IHC for MMR protein, p53, and L1CAM. Subsequently, perform a POLE test only when any one of the following criteria is positive: p53 abnormal, L1CAM positive, stage IB-II, high grade, substantial LVSI.”
Line 405-409: “Despite the limitations of being a single-institution retrospective study with a relatively small sample size, our findings indicate the potential benefit of integrating POLE by ddPCR and L1CAM IHC into the current risk-stratification approach. We acknowledge the need for further validation through large-scale, prospective multicenter studies to confirm the utility of our proposed classification method.”
: Line 397-399: “There are several limitations in this study: 1) it was a retrospective study, 2) conducted in a single institution, and 3) involved a relatively small sample size, particularly for cases of the NSMP-L1CAMpos subtype. Therefore, further validation through a large-scale, prospective multicenter study is necessary to confirm and generalize our findings.”
Round 2
Reviewer 2 Report
I have no further comments